# Empirical Likelihood for Contextual Bandits

**Nikos Karampatziakis**
Microsoft Dynamics 365 AI
nikosk@microsoft.com

**John Langford**
Microsoft Research
jcl@microsoft.com

**Paul Mineiro**
Microsoft Research
pmineiro@microsoft.com

## Abstract

We propose an estimator and confidence interval for computing the value of a policy from off-policy data in the contextual bandit setting. To this end we apply empirical likelihood techniques to formulate our estimator and confidence interval as simple convex optimization problems. Using the lower bound of our confidence interval, we then propose an off-policy policy optimization algorithm that searches for policies with large reward lower bound. We empirically find that both our estimator and confidence interval improve over previous proposals in finite sample regimes. Finally, the policy optimization algorithm we propose outperforms a strong baseline system for learning from off-policy data.

## 1 Introduction

Contextual Bandits [3, 17] are now in widespread practical use ([19, 7, 25]). Key to their success is the ability to do *off-policy* or *counterfactual estimation* [12] of the value of any policy enabling sound train/test regimes similar to supervised learning. However, off-policy evaluation requires more data than supervised learning to produce estimates of the same accuracy. This is because off-policy data needs to be importance-weighted and accurate estimation for importance-weighted data is still an active research area. How can we find a tight confidence interval (CI) on counterfactual estimates? And since tight CIs are deeply dependent on the form of their estimate, how can we find a tight estimate? And given what we discover, how can we leverage this for improved learning algorithms?

We discover good answers to these questions through the application of empirical likelihood [24], a nonparametric maximum likelihood approach that treats the sample as a realization from a multinomial distribution with an infinite number of categories. Like a likelihood method, empirical likelihood (EL) adapts to the difficulty of the problem in an automatic way and results in efficient estimators. Unlike parametric likelihood methods, we do not need to make any parametric assumptions about the data generating process. We do assume that the expected importance weight is 1, a nonparametric moment condition that is supposed to hold for correctly collected off-policy data. Finally, EL-based estimators and confidence intervals can be computed by efficient algorithms that solve low dimensional convex optimization problems. Figure 1 shows a preview of our results.

In section 4.2 we introduce our estimator. The estimator is computationally tractable, requiring a bisection search over a single scalar, has provably low bias (see Theorem 1) and in section 5.1 we experimentally demonstrate performance exceeding that of popular alternatives.

The estimator leads to an asymptotically exact confidence interval for off-policy estimation which we describe in section 4.3. Other CIs are either narrow but fail to guarantee prescribed coverage, or guarantee prescribed coverage but are too wide to be useful. Our interval is narrow and (despite having only an asymptotic guarantee) empirically approaches nominal coverage from above as in Figure 1 and Table 3. Finally, in section 4.5, we use our CI to construct a robust counterfactual learning objective. We experiment with this in section 5.3 and empirically outperform a strong baseline.

We now highlight several innovations in our approach:

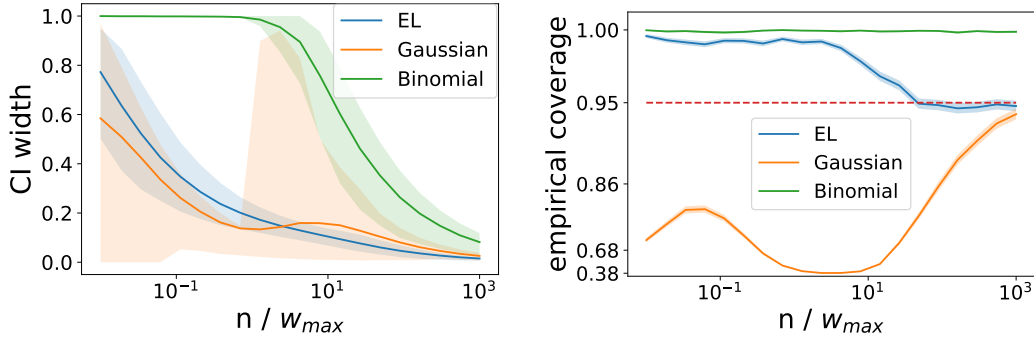

Figure 1: A comparison of confidence intervals on contextual bandit data. The EL confidence interval is dramatically tighter than an approach based on a binomial confidence interval while avoiding chronic undercoverage as per the asymptotic Gaussian confidence interval. In some regimes, the asymptotic Gaussian CI both undercovers *and* has greater average width. This is possible as the EL CI has a different functional form than a multiplier on the Gaussian CI. On the left, shaded area represents 90% of the empirical distribution indicating the EL CI width varies less over realizations. On the right, shaded area represents 4 times the standard error of the mean indicating coverage differences are everywhere statistically significant.

- We use a nonparametric likelihood approach. This maintains [15] some of the asymptotic optimality results known for likelihood in the multinomial (hence well-specified) case[11].

- We prove a finite sample result on the bias of our estimator. This also implies our estimator is asymptotically consistent.

- Our CI considers a large set of plausible worlds (alternative hypotheses) from which the observed off-policy data could have come from. One implication (cf. section 4.4) is that for binary rewards the CI lower bound will be $< 1$ (and $> 0$) even if all observed rewards are $1$.

- We show how to compute the confidence interval directly, saving a factor of $\log(1/\epsilon)$ in time complexity compared to standard implementations of EL for general settings.

- We propose a learning objective that searches for a policy with the best lower bound on its reward and draw connections with distributionally robust optimization.

## 2   Related Work

There are many off-policy estimators for contextual bandits. The "Inverse Propensity Score" (IPS) [12] is unbiased, but has high variance. The Self-Normalized IPS (SNIPS) [30] estimator trades off some bias for better mean squared error (MSE). Our estimator has bias of the same order as SNIPS and empirically better MSE. The EMP estimator of [14] also uses EL techniques and we will explain the differences in detail in section 4.2. Critically, it would be challenging to use EMP to construct a CI with correct coverage for small samples, as we will explain in section 4.4. An orthogonal way to reduce variance is to incorporate a reward estimator as in the doubly robust (DR) estimator and associated variants [27, 9, 33, 32]. The estimator presented here is a natural alternative to IPS and SNIPS and can naturally replace the IPS part of a doubly robust estimator.

There is less work on off-policy CIs for contextual bandits. A simple baseline randomly rounds the rewards to $\{0, 1\}$ and the importance weights to $0$ or the largest possible weight value and applies a Binomial confidence interval. Another simple asymptotically motivated approach, previously applied to contextual bandits [18], is via a Gaussian approximation. The EL confidence intervals are also asymptotically motivated but empirically approach nominal coverage from above and are much tighter than the Binomial confidence interval. In [5] empirical Bernstein bounds or Gaussian approximations are combined with clipping of large importance weights to trade bias for variance. This requires hyperparameter tuning whereas EL provides parameter-free CIs. Similar ideas to ours have been used for the upper confidence bound in the Empirical KL-UCB algorithm [6], an on-policy algorithm for multi-armed bandits. As detailed in section 4.4, both constructions need to consider some events that

may not be in the data. While this happens without explicit data augmentation, it is analogous to the use of explicitly augmented MDPs for off-policy estimation in Markov Decision Processes[20].

Learning algorithms for contextual bandits include theoretical [3, 17], reduction oriented [9], optimization-based [29], and Bayesian [21] algorithms. A recent paper about empirical contextual bandit learning [4] informs our experiments.

Ideas from empirical likelihood have previously been applied to robust supervised learning [8]. Our combination of CIs with learning is a contextual bandit analogue to robust supervised learning. Regularizing counterfactual learning via lower-bound optimization has been previously considered, e.g., based upon empirical Bernstein bounds [29] or divergence-based trust regions grounded in lower bounds from conservative policy iteration [28, 13].

## 3 Notation and Warm-up

We consider the off-policy contextual bandit problem, with contexts $x \in \mathcal{X}$, a finite set of actions $A$, and bounded real rewards $\mathbf{r} \in A \to [0, 1]$. The environment generates i.i.d. context-reward pairs $(x, \mathbf{r}) \sim D$ and first reveals $x$. Then an action $a \in A$ is sampled and the reward $\mathbf{r}(a)$ is revealed.

Let $\pi$ be the policy whose value we want to estimate. For off-policy estimation we assume a dataset $\{(x_n, a_n, p_n, r_n)\}_{n=1}^N$, generated from an arbitrary sequence of historical stochastic policies $h_n$, with $p_n \doteq h_n(a_n|x_n)$ and $r_n \doteq \mathbf{r}_n(a_n)$. Let $w(a) \doteq \frac{\pi(a|x)}{h(a|x)}$ be a random variable denoting the density ratio between $\pi$ and $h$ and $w_n \doteq \frac{\pi(a_n|x_n)}{h_n(a_n|x_n)}$ its realization. We assume $\pi \ll h_n$ (absolute continuity), and that $w \in [w_{\min}, w_{\max}]$.[1] The value of $\pi$ is defined as $V(\pi) = \mathbb{E}_{(x,\mathbf{r}) \sim D, a \sim \pi(\cdot|x)}[\mathbf{r}(a)]$. Since we don't have data from $\pi$, but from $h_n$ we use importance weighting to write $V(\pi) = \mathbb{E}_{(x,\mathbf{r}) \sim D, a \sim h(\cdot|x)}[w(a)\mathbf{r}(a)]$. The inverse propensity score (IPS) estimator is a direct implementation of this: $V^{\text{IPS}}(\pi) = \frac{1}{N} \sum_{n=1}^N w_n r_n$. We can do better by observing that each policy $h_n$ is created using data before time $n$. Formally, let $\{\mathcal{F}_n\}$ be the filtration generated by $\{(x_k, a_k, p_k, r_k)\}_{k<n}$, and assume $\{h_n\}$ is $\{\mathcal{F}_n\}$-adapted. Let $\mathbb{E}_n[\cdot] \doteq \mathbb{E}[\cdot|\mathcal{F}_n]$. These observations allow us to note that $\forall n : \mathbb{E}_n[w(a)] = 1$. This moment condition has been used for variance reduction (e.g in the SNIPS estimator). We also observe that $m_n(v) = \begin{pmatrix} \sum_{k \leq n} (w_k r_k - v) \\ \sum_{k \leq n} (w_k - 1) \end{pmatrix}$ is a martingale sequence when $v = V(\pi)$. This observation will allow us to develop consistent estimators even with a non-stationary behavior policy.

### 3.1 Pedagogical Example

Suppose $\pi$ is deterministic, $\mathbf{r}$ is binary-valued, and $h_n$ is the same $\epsilon$-greedy policy for all $n$. In this case there are only 3 possible values[2] for the importance weight $w_n = \frac{\pi(a_n|x_n)}{h_n(a_n|x_n)}$; 2 possible values for the reward, and the data is an i.i.d. sample. The observed data can be reduced to a histogram with 6 bins. To construct an estimator and a confidence interval we will reason about plausible worlds that could have generated the data. In particular each of these worlds induces a joint distribution over importance weights and rewards. Let $Q^*_{w,r}$ denote the true probability of $(w, r)$ under the logging policy. Its maximum likelihood estimator is

$$Q^{\text{mle}} = \arg\max_{Q \in \Delta} \left\{ \sum_n \log (Q_{w_n, r_n}) \middle| \mathbb{E}_Q[w] = 1 \right\},$$

where $\Delta$ is the simplex. The constraint enforces that the counterfactual distribution under $\pi$ normalizes; we discuss the implications in section 4.4. Associated with any maximizer $Q^{\text{mle}}$ is a corresponding value estimate $\hat{V}(\pi) = \mathbb{E}_{Q^{\text{mle}}}[wr]$. Absent the constraint, the maximizing $Q$ would have been the empirical distribution and $\hat{V}(\pi)$ would be $V^{\text{IPS}}(\pi)$. Furthermore, to find an asymptotic

CI for $\hat{V}(\pi)$, we can use Wilks' theorem. Define the maximum profile likelihood at $v$:

$$L(v) = \sup_{Q \in \Delta} \left\{ \sum_n \log\left(Q_{w_n,r_n}\right) \middle| \mathbb{E}_Q[w] = 1, \mathbb{E}_Q[wr] = v \right\}. \tag{1}$$

Let $Q^{\text{prof}}(v)$ be the maximizing $Q$ for $L(v)$. Wilks' Theorem says that $-2(L(V(\pi)) - \sum_n \log(Q^{\text{mle}}_{w_n,r_n})) \to \chi^2_{(1)}$ in distribution as $n \to \infty$. Letting $\chi^{2,1-\alpha}_{(1)}$ be the $1-\alpha$-quantile of a $\chi$-square distribution with one degree of freedom, an asymptotic $1-\alpha$-confidence interval is

$$\left\{ v \middle| \sum_n \log(Q^{\text{mle}}_{w_n,r_n}) - \sum_n \log(Q^{\text{prof}}_{w_n,r_n}(v)) \leq \frac{1}{2}\chi^{2,1-\alpha}_{(1)} \right\}.$$

That is, if for a candidate $v$ there exists a distribution $Q^{\text{prof}}(v)$ over $(w, r)$ pairs such that $\mathbb{E}_{Q^{\text{prof}}(v)}[w] = 1$, $\mathbb{E}_{Q^{\text{prof}}(v)}[wr] = v$, and the data likelihood is high then $v$ should be in the CI for $V(\pi)$. [11] shows that for multinomials this is the tightest $1-\alpha$-confidence interval as $n \to \infty$ and $\alpha \to 0$.

The value estimate is not necessarily unique if there are multiple distributions $Q^{\text{mle}}$ which obtain the maximum, but all value estimates are contained in the $\alpha \to 1$ limit of the above CI. For instance, if all observed importance weights are zero by chance, $Q^{\text{mle}}$ must place some mass on a $(w, r)$ with $w > 1$ to satisfy $\mathbb{E}_Q[w] = 1$, but the likelihood is not sensitive to the value of $r$.

# 4 Off-Policy Estimation and Confidence Interval

We first review how empirical likelihood extends the above results, then present our results.

## 4.1 Empirical Likelihood

So far, we assumed that the random vector $(w, r)$ has finite support and that data is iid. Empirical likelihood [24] allows us to transfer the above results to settings where the support is infinite. The seminal work [23] showed that the finite support assumption is immaterial. This was later extended [26] to prove that estimating equations such as $\mathbb{E}[w] = 1$ could be incorporated into the estimation of $\mathbb{E}[wr]$. As long as $\text{Cov}(w, wr) \neq 0$ Corollary 5 of [26] (also Theorem 3.5 of [24]) implies that $-2(L(V(\pi)) - \sum_n \log(Q^{\text{mle}}_{w_n,r_n})) \to \chi^2_{(1)}$ in distribution as $n \to \infty$ without assuming finite support for $(w, r)$. Asymptotic optimality results for empirical likelihood are established in [15], but require different proof techniques from the multinomial case [11].

We now turn to the iid. assumption. In many practical setups data may have been collected from various logging policies which makes the $w$'s non-iid. Existing estimators, such as IPS, have no trouble handling such data. A key insight is that all the information about the problem is captured in the martingale estimating equation $m_n(V(\pi)) = 0$. The extension of empirical likelihood to martingales is given by Dual Likelihood [22]. The reason for the name is that the functional of interest is the convex dual of the empirical likelihood formulation subject to the martingale estimating equation of interest. In our case, we use dual variables $\tau$ and $\beta$ that correspond to the first and second component of $m_n(v) = 0$ respectively. As derived in appendix A we get the dual likelihood

$$l_v(\beta, \tau) = \sum_n \log\left(1 + \beta(w_n - 1) + \tau(w_n r_n - v)\right) \tag{2}$$

That derivation also reveals the constraint set associated with a feasible primal solution,

$$\mathcal{C} = \{(\beta, \tau) | \forall w, r : 1 + \beta(w - 1) + \tau(wr - v) \geq 0\}. \tag{3}$$

Despite the domains of $w$ and $r$ being potentially infinite, we can express $\mathcal{C}$ using only 4 constraints as $\mathcal{C} = \{(\beta, \tau) | \forall w \in \{w_{\min}, w_{\max}\}, r \in \{0, 1\} : 1 + \beta(w - 1) + \tau(wr - v) \geq 0\}$.

This is also the convex dual of (1) as iid and finite support data are just special cases of this framework. However, $L(v)$ and the corresponding $Q$ do not have a generative interpretation when $w$'s are not iid. Nevertheless, under very mild conditions [22] the maximum of eq. (2) with $v = V(\pi)$ still has an asymptotic distribution that obeys a nonparametric analogue to Wilks' theorem. Thus it functions similarly for hypothesis testing. We will still refer to the support of $Q$ to provide intuition.

What is the set of alternative hypotheses considered when constructing hypothesis tests or CIs via a dual likelihood formulation? This is easier to understand in the primal, as the dual likelihood corresponds to a primal optimization over all distributions $Q$ over $(w, r)$ which measure-theoretically dominate the empirical distribution (i.e., place positive probability on each realized datum) and satisfy the moment condition $\mathbb{E}_Q[w] = 1$. Although this includes distributions with unbounded support, the optima are supported on the sample plus at most one more point as discussed in section 4.4.

## 4.2 Off-Policy Estimation

We start by defining a (dual) analogue to the nonparametric maximum likelihood estimator (NPMLE) in the primal formulation for the iid case. Consider the quantity

$$l^*_{\text{mle}} = \sup_{(\beta, 0) \in \mathcal{C}} l_v(\beta, 0) \tag{4}$$

which is obtained by setting $\tau = 0$ (so the value of $v$ is immaterial) and optimizing over $\beta$. This quantity may seem mysterious, but it corresponds to the NPMLE. Indeed, $\tau = 0$ means $\mathbb{E}_Q[wr]$ is free to take on any value, as in the primal maximum likelihood formulation. We propose our estimator as any $v$ which obtains the maximum dual likelihood, i.e., any value in the set

$$\left\{ v \,\middle|\, \sup_{(\beta, \tau) \in \mathcal{C}} l_v(\beta, \tau) = l^*_{\text{mle}} \right\}. \tag{5}$$

In appendix B we prove there is an interval of maximizers of the form

$$\hat{V}(\pi; \rho) = \rho + \frac{1}{N} \sum_n \frac{w_n(r_n - \rho)}{1 + \beta^*(w_n - 1)}, \tag{6}$$

where $\rho$ is any value in $[0, 1]$ and $\beta^*$ maximizes

$$\sum_n \log\left(1 + \beta(w_n - 1)\right) \text{ s.t. } \forall w : 1 + \beta(w - 1) \geq 0. \tag{7}$$

The constraints on $\beta^*$ are over all possible values of $w$, not just the observed $w$. However the constraints with $w = w_{\min}$ and $w = w_{\max}$ imply all other constraints. We solve this 1-d convex problem via bisection to accuracy $\epsilon$ in $O(N \log(\frac{1}{\epsilon}))$ time. Note that $\beta = 0$ is always feasible and it is optimal when $\sum_n w_n = N$. When $\beta^* = 0$, (6) becomes $V^{\text{IPS}}$ for all values of $\rho$.

Eq. (6) (and eq. (9) in 4.3) are valid in the martingale setting, i.e., for a sequence of historical policies. Appendix B shows that when there exists an unobserved extreme value of $w$, say $w_{ex}$, any associated primal solution $Q^{\text{mle}}$ will assign some probability to a pair $(w_{ex}, \rho)$. Section 4.4 discusses the beneficial implications of this. Once both $w_{\min}, w_{\max}$ are observed with any $r$, eq. (6) becomes a point estimate because $\sum_n w_n (1 + \beta^*(w_n - 1))^{-1} = N$, i.e., $\rho$ cancels out and $Q^{\text{mle}}$ only has support on the observed data.

The EMP estimator, based on empirical likelihood, was proposed in [14]. Specializing it to a constant reward predictor for all $(x, a)$ we can write both estimators in terms of $Q^{\text{mle}}$. Eq. (6) leads to $\hat{V}(\pi) = (1 - \sum_n Q^{\text{mle}}_{w_n, r_n} w_n)\rho + \sum_n Q^{\text{mle}}_{w_n, r_n} w_n r_n$ while EMP is $\hat{V}_{\text{EMP}}(\pi) = \sum_n Q^{\text{mle}}_{w_n, r_n} w_n r_n / \sum_n Q^{\text{mle}}_{w_n, r_n}$. When $w_{\min}$ and $w_{\max}$ are observed, $\sum_n Q^{\text{mle}}_{w_n, r_n} = \sum_n Q^{\text{mle}}_{w_n, r_n} w_n = 1$ and the two estimators coincide. Section 5.1 empirically investigates their finite sample behavior.

### 4.2.1 Finite Sample Bias

We show a finite-sample bound on the bias of an estimator, based upon eq. (6), of the value difference $R(\pi) \doteq V(\pi) - V(h)$ between $\pi$ and the logging policy. We obtain our estimator for $R(\pi)$ via $\mathbb{E}_{Q^{\text{mle}}}[wr] - \mathbb{E}_{Q^{\text{mle}}}[r]$ and using the primal-dual relationship for $Q^{\text{mle}}$ from appendix A. In practical applications $R(\pi)$ is the relevant quantity for deciding when to update a production policy. The proof is in appendix D.

**Theorem 1.** *Let $\hat{R}(\pi) \doteq \frac{1}{N} \sum_n \frac{(w_n - 1)(r_n - \rho)}{1 + \beta^*(w_n - 1)}$ with $\beta^*$ as in eq. (7), and let a.s. $\forall n : 0 \leq w_n \leq w_{\max}$ with $w_{\max} \geq 1$. Then*

$$\left| \mathbb{E}\left[\hat{R}(\pi)\right] - R(\pi) \right| \leq 10\sqrt{\frac{w_{\max}}{N}} + 16\frac{w_{\max}}{N}$$

*where $R(\pi) \doteq V(\pi) - V(h)$ is the true policy value difference between $\pi$ and $\{h_n\}_{n \in N}$.*

The leading term in Theorem 1 is actually any $\omega \geq \mathbb{E}_n[(w_n - 1)^2]$; $\omega = w_{\max}$ is a worst case. This result indicates low bias for the estimator (leading terms comparable to finite-sample variance); meanwhile inspection of eq. (6) (and (9)) indicate neither can overflow the underlying range of reward. This explains the excellent mean square error performance observed in section 5.1.

## 4.3   Off-Policy Confidence Interval

We can use the dual likelihood of eq. (2) to construct an asymptotic confidence interval [22] in a manner completely analogous to Wilks' theorem for the primal likelihood formulation

$$\left\{ v \, \middle| \, \sup_{(\beta,\tau) \in \mathcal{C}} l_v(\beta,\tau) - l^*_{\text{mle}} \leq \frac{1}{2} \chi^{2,\alpha}_{(1)} \right\}, \tag{8}$$

where $\alpha$ is the desired nominal coverage and $\chi^{2,\alpha}_{(1)}$ is the $\alpha$-quantile of a $\chi$-square distribution with one degree of freedom. The asymptotic guarantee is that the coverage error of this interval is $O(1/n)$.

In general applications of EL a bisection on $v$ is recommended for finding the boundaries of the CI: given an interval $[\ell, u]$ check whether $v = (\ell + u)/2$ is in the set given by (8) and update $\ell$ or $u$. This requires $O(\log(1/\epsilon))$ calls to maximize (2). Here we derive a more explicit form for the boundary points which is more insightful and faster to compute (2 optimization calls). In appendix C we prove the lower bound of the CI is

$$v_{\text{lb}}(\pi) = \kappa^* \frac{1}{N} \sum_n \frac{w_n r_n}{\gamma^* + \beta^* w_n + w_n r_n}, \tag{9}$$

where $(\beta^*, \gamma^*, \kappa^*)$ are given by

$$\sup_{\substack{\kappa \geq 0 \\ \beta, \gamma}} \sum_n \left( -\kappa \log \kappa + \kappa \left( -\phi + \log \left( \gamma + \beta w_n + w_n r_n \right) \right) \right)$$

subject to $\forall w : \gamma + \beta w \geq 0$, where $\phi = \frac{1}{2N} \chi^{2,\alpha}_{(1)} - \frac{1}{N} l^*_{\text{mle}}$.

The constraints range over all possible values of $w$, but $w_{\min}$ and $w_{\max}$ are the only relevant ones. This is a convex problem with 3 variables and 2 constraints that can be solved to $\epsilon$-accuracy by the ellipsoid method (for example) in $O(N \log(\frac{1}{\epsilon}))$ time. The upper bound can be obtained by transforming the rewards $r \leftarrow 1 - r$, finding the lower bound, and then setting $v_{\text{ub}} \leftarrow 1 - v_{\text{lb}}$.

In eq. (9) we can have $\kappa^* \sum_n w_n \left( \gamma^* + \beta^*(w_n - 1) + w_n r_n \right)^{-1} < N$ even after extreme values of $w$ have been observed. This corresponds to a primal solution which is placing "extra" probability on either $(w_{\min}, 0)$ or $(w_{\max}, 0)$. For example, this allows our lower bound to be $< 1$ even if all observed rewards are 1. Section 4.4 discusses the benefits of the additional primal support.

## 4.4   The Importance of $\mathbb{E}[w] = 1$

By inspection, the primal constraint $\mathbb{E}[w] = 1$ can be infeasible for a distribution only supported on the observed values if 1 is not in the convex hull of observed importance weights. Consequently solutions to equations (6) and (9) can correspond to distributions $Q$ in the primal formulation with support beyond the observed values. This is a known property of constrained empirical likelihood [10].

We precisely characterize the additional support as confined to a single extreme point. In appendix B we show the support of the primal distribution associated with equation (6) is a subset of $\{(w_n, r_n)\}_{n \leq N} \cup \{(w_{ex}, \rho)\}$, where $w_{ex} = w_{\min}$ if $\sum w_n \geq N$ and otherwise $w_{ex} = w_{\max}$. In appendix C we show the support of the primal distribution associated with equation (9) is $\{(w_n, r_n)\}_{n \leq N} \cup \{(w_{ex}, 0)\}$, where $w_{ex}$ is either $w_{\min}$ or $w_{\max}$. We also point out that similar ideas have already been used for multi-armed bandits. For example, the empirical KL-UCB algorithm [6] uses empirical likelihood to construct an upper confidence bound on each arm by considering distributions that can place additional mass on the largest possible reward.

Although the modification of the support from the observed data points seems modest, it greatly improves both the estimator and the CI. Critically, both can produce values that are outside the convex

hull of the observations, but never overflow the possible range $[0, 1]$. In contrast, empirical likelihood on the sample is constrained to the convex hull of the observations; while empirical likelihood on the bounded range without the $\mathbb{E}[w] = 1$ constraint can produce value estimates in the range $[0, w_{\max}]$. Furthermore, we observe in practice that our CIs approach nominal coverage values from above, as in Figure 1. This is not typical behavior when empirical likelihood is constrained to the sample.

Per Lemma 2.1 of [24], empirical likelihood can only place $O(1/n)$ mass outside the sample. With our primal constraint $\mathbb{E}[w] = 1$ this mass is further limited to $O(1/w_{\max})$, and decreases as the realized average importance weight approaches 1. As seen in Figure 1, this can result in non-trivial CIs in the regime $n < w_{\max}$ where other interval estimation techniques struggle.

### 4.5 Offline Contextual Bandit Learning

Here the goal is to learn a policy $\pi$ using a dataset $\{(x_n, a_n, p_n, r_n)\}_{n \in N}$, i.e., without interacting with the system generating the data. One strategy is to leverage a counterfactual estimator to reduce policy learning to optimization [18], suggesting the use of equation (6) in the objective.

Alternatively we can instead optimize the lower bound of equation (9). In the iid. case optimizing the lower bound corresponds to a variant of distributionally robust optimization. The log-empirical likelihood for a distribution $Q$ is equivalent to the KL divergence between the empirical distribution $\mathbb{1}/N$ and $Q$. A likelihood maximizer $Q^{\text{mle}}$ attains the minimum such KL divergence. By optimizing the lower bound (9) we are performing distributionally robust optimization with uncertainty set

$$ \mathcal{Q}(\pi) = \left\{ Q \middle| \mathbb{E}_Q[w(\pi)] = 1, \ \text{KL}\left( \frac{\mathbb{1}}{N} \middle\| Q \right) \le B(\pi) \right\}, $$

where $B(\pi) = \text{KL}\left( \frac{\mathbb{1}}{N} \| Q^{\text{mle}}(\pi) \right) + \frac{1}{2N} \chi^{2,\alpha}_{(1)}$ and we have made dependences on $\pi$ explicit. Given a set of policies $\Pi$ we can set up the game

$$ \max_{\pi \in \Pi} \min_{Q \in \mathcal{Q}(\pi)} \sum_n Q_{w(\pi)_n, r_n} w(\pi)_n r_n $$

for finding the policy $\pi^* \in \Pi$ with the best reward lower bound. For our experiments we use a heuristic alternating optimization strategy. In one phase the policy is fixed and we find the optimal dual variables associated with equation (9). In the alternate phase we find a policy with a better lower bound, i.e., a policy which improves upon equation (9) with dual variables held fixed. Developing better methods for solving this game is deferred for future work.

## 5 Experiments

The purpose of our experiments is to demonstrate the empirical behavior of the proposed methods against other methods that use the same information. Comparing against methods that leverage or focus on reward predictors is therefore out of scope, as reward predictors can help/hurt any method. Our experiments compare MSE of estimators (section 5.1), confidence interval coverage and width (section 5.2), and utility of lower bound optimization for off-policy learning (section 5.3).

Replication instructions are available in the supplement, and replication software is available at http://github.com/pmineiro/elfcb. All experiment details are in the appendix.

### 5.1 Off-Policy Estimation

**Synthetic Data** We begin with a synthetic example to build intuition. In appendix E we detail how we sample $w = \pi/h$ and $r$ for each synthetic environment. Figure 2 shows the mean squared error (MSE) over 10,000 environment samples for various estimators. The best constant predictor of 1/2 ("Constant") has a MSE of 1/12, as expected. ClippedDR is the doubly robust estimator with the best constant predictor of 1/2 clipped to the range $[0, 1]$, i.e. $\min(1, \max(0, \frac{1}{2} + \sum_n \frac{w_n}{N}(r_n - 1/2)))$. SNIPS is the self-normalized estimator IPS estimator. EMP is the estimator of [14]. For EL, we use $\rho = \frac{1}{2}$. When a small number of large importance weight events is expected in a realization, both ClippedDR and SNIPS suffer due to their poor handling of the $\mathbb{E}[w] = 1$ constraint. EMP is an improvement and EL is a further improvement. Asymptotically all estimators are similar.

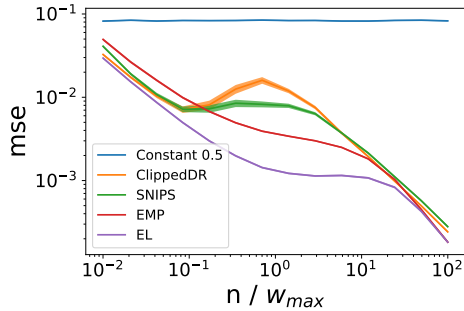

Figure 2: Mean squared error of EL and other estimators on synthetic data. Asymptotics are similar while EL dominates in the small sample regime. Line width is 4 times the standard error of the population mean.

| EL vs. | Exploration | Wins | Ties | Losses |
|---|---|---|---|---|
| IPS | $\epsilon = 0.05$ | 26 | 11 | 3 |
|  | bags=10 | 13 | 19 | 8 |
|  | cover=10 | 16 | 16 | 9 |
| SNIPS | $\epsilon = 0.05$ | 5 | 34 | 1 |
|  | bags=10 | 7 | 30 | 3 |
|  | cover=10 | 7 | 33 | 0 |
| EMP | $\epsilon = 0.05$ | 24 | 13 | 3 |
|  | bags=10 | 8 | 26 | 6 |
|  | cover=10 | 8 | 23 | 9 |

Table 1: Off-policy evaluation results where $\epsilon = 0.05$ is $\epsilon$-greedy exploration, bags=10 is bootstrap exploration with 10 replicas, and cover=10 is online cover [2] with 10 policies.

| Exploration | CI LB | | | EL | | |
|---|---|---|---|---|---|---|
|  | Wins | Ties | Losses | Wins | Ties | Losses |
| $\epsilon = 0.05$ greedy | 16 | 18 | 6 | 11 | 26 | 3 |
| $\epsilon = 0.1$ greedy | 16 | 19 | 5 | 13 | 24 | 3 |
| $\epsilon = 0.25$ greedy | 15 | 22 | 3 | 3 | 34 | 3 |
| bagging, 10 bags | 21 | 18 | 1 | 11 | 28 | 1 |
| bagging, 32 bags | 4 | 26 | 10 | 7 | 31 | 2 |
| cover, 10 policies | 18 | 21 | 1 | 6 | 30 | 4 |
| cover, 32 policies | 9 | 29 | 2 | 6 | 34 | 0 |

Table 2: Learning results. "CI LB" uses equation (9); "EL" uses equation (6). "EL" serves as an ablation study, on whether the improvement in "CI LB" is due to distributional robustness, or the estimator itself.

**Realistic Data** We employ an experimental protocol inspired by the operations of the Decision Service [1], an industrial contextual bandit platform. Details are in appendix F. Succinctly, we use 40 classification datasets from OpenML [31]; apply a supervised-to-bandit transform [9]; and limit the datasets to 10,000 examples. Each dataset is randomly split 20%/60%/20% into Initialize/Learn/Evaluate subsets, to learn $h$, learn $\pi$, and evaluate $\pi$ respectively. Learning is via Vowpal Wabbit [16] using various exploration strategies, with default parameters and $\pi$ initialized to $h$.

We compare the MSE of EL, IPS, SNIPS, and EMP using the true value of $\pi$ on the evaluation set (available because the underlying dataset is fully observed and $\pi(a|x)$ is known). For each dataset we evaluate multiple times, each time resampling $a \sim h(\cdot|x)$. Table 1 shows the results of a paired $t$-test with 60 trials per dataset and 95% confidence level: "tie" indicates null result, and "win" or "loss" indicates significantly better or worse. The EL is overall superior to IPS and SNIPS. It is similar to EMP except when the data comes from 0.05-greedy exploration, where EL is better than EMP.

## 5.2 Confidence Intervals

**Synthetic Data** We use the same synthetic $\epsilon$-greedy data as described above. Figure 1 shows the mean width and empirical coverage over 10,000 environment samples for various CIs at 95% nominal coverage. Binomial CI is the Clopper Pearson confidence interval on the random variable $\frac{w}{w_{\max}}R$. This is an excessively wide CI. Asymptotic Gaussian is the standard z-score CI around the empirical mean and standard deviation motivated by the central limit theorem. Intervals are

Table 3: Off-Policy Confidence Intervals

| Technique | Coverage (Average) | Width Ratio (Median) |
|---|---|---|
| EL | 0.975 | n/a |
| Binomial | 0.996 | 2.89 |
| AG | 0.912 | 0.99 |

narrow but typically violate nominal coverage. The EL interval is narrow and obeys nominal coverage throughout the entire range despite only having asymptotic guarantees.

Once again there is a qualitative change when the sample size is comparable to the largest importance weight. The Binomial CI interval only begins to make progress at this point. Meanwhile, the asymptotic Gaussian interval widens as empirical variance increases.

**Realistic Data** We use the same datasets mentioned above, but produce a 95% confidence interval for off-policy evaluation rather than the maximum likelihood estimate. With 40 datasets and 60 evaluations per dataset, we have 2400 confidence intervals from which we compute the coverage and the ratio of the width of the interval to the EL in table 3. As expected from simulation, the Binomial CI overcovers and has wider intervals. EL widths are comparable to asymptotic Gaussian (AG) on this data, but AG undercovers. A 95% binomial confidence interval on the coverage of AG is $[90.0\%, 92.3\%]$, indicating sufficient data to conclude undercoverage.

### 5.3 Offline Contextual Bandit Learning

We use the same 40 datasets as above, but with a 20%/20%/60% Initialize/Learn/Evaluate split. We made no effort to tune the confidence level setting it to 95% for all experiments. For optimizing the policy parameters and the distribution dual variables, we alternate between solving the dual problem with the policy fixed and then optimizing the policy with the dual variables fixed. To optimize the policy we do a single pass over the data using Vowpal Wabbit as a black-box oracle for learning, supplying different importance weights on each example depending upon the dual variables. We do 4 passes over the learning set and update the dual variables before each pass. Details are in appendix G.

We compare the true value of $\pi$ on the evaluation set resulting from learning with the different objectives. For each dataset we learn multiple times, with different actions chosen by the historical policy $h$. Table 2 shows the results of a paired $t$-test with 60 trials per dataset and 95% confidence level: "tie" indicates null result, and "win" or "loss" indicates significantly better or worse evaluation value for the CI lower bound. Using the CI lower bound overall yields superior results. Using the EL estimate also provides some lift but is less effective than using the CI lower bound.

## 6 Conclusions

We presented a practical estimator and a CI for contextual bandits with correct asymptotic coverage and empirically valid coverage for small samples. To this end we used empirical likelihood techniques which yielded computationally efficient and hyperparameter-free procedures for estimation, CIs and learning. Empirically, our proposed CI is a substantial improvement over existing methods and the learning algorithm is a useful improvement against techniques that optimize the value of a point estimate. Our methods offer the largest advantage in regimes where existing methods struggle, such as when the number of samples $N$ is of the same order as the largest possible importance weight.

## Broader Impact

Not applicable to this work.

## Acknowledgments and Disclosure of Funding

We thank Adith Swaminathan and the anonymous reviewers for their valuable comments on earlier drafts on this work.

## Footnotes

[1] $w_{\min} = 0$ is always a lower bound, but $w_{\max}$ is application dependent. To ensure $\pi \ll h_n$ so that estimation is consistent, it is common to enforce, for every action $a'$, $h_n(a'|x_n) \geq p_{\min}$. Then $w_{\max} \leq 1/p_{\min}$.

[2] $\pi(a_n|x_n) \in \{0, 1\}$ and $h_n(a_n|x_n)$ has two possible values.

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
