[Supplementary Material]

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

## A  Derivation of Profile Likelihood

For ease of exposition, we will start with a primal formulation and via duality show equivalence with Dual Likelihood [22] applied to the Doléans-Dade multiplicative martingale corresponding to $m_n(v)$.

Starting from

$$\sup_{Q\in\Delta}\left\{\sum_n \log\left(Q_{w_n,r_n}\right)\middle| \mathbb{E}_Q[w]=1, \mathbb{E}_Q[wr]=v\right\}.$$

we form the Lagrangian dual

$$\sup_{\beta,\gamma,\tau}\inf_{Q\succeq 0}\left\{\beta\left(-1+\sum_{w,r}wQ_{w,r}\right)+\gamma\left(-1+\sum_{w,r}Q_{w,r}\right)\right.$$

$$\left.+\tau\left(-v+\sum_{w,r}wrQ_{w,r}\right)-\sum_{w,r}c_{w,r}\log\left(Q_{w,r}\right)\right\},$$

where $c_{w,r}=\sum_n 1_{w=w_n,r=r_n}$. Collecting terms

$$\sup_{\beta,\gamma,\tau}\inf_{Q\succeq 0}\left\{-\beta-\gamma-\tau v\right.$$

$$\left.+\sum_{w,r}\left(\beta w+\gamma+\tau wr\right)Q_{w,r}-c_{w,r}\log\left(Q_{w,r}\right)\right\}.$$

Dual boundedness requires $\forall w,r: \beta w+\gamma+\tau wr\geq 0$. The infimum over $Q$ is separable yielding

$$Q^*_{w,r}=\frac{c_{w,r}}{\beta w+\gamma+\tau wr},$$

if $c_{w,r}>0$ or $\beta w+\gamma+\tau wr>0$, otherwise the contribution to the dual is zero. Substituting

$$\sup_{\beta,\gamma,\tau}\left\{-\beta-\gamma-\tau v+\sum_n \log\left(\beta w_n+\gamma\right)\right.$$

$$\left.\middle|\forall w: \beta w+\gamma+\tau wr\geq 0\right\}$$

discarding constants.

Summing the KKT stationarity conditions yields

$$\frac{c_{w,r}}{Q_{w,r}}=\beta w+\gamma+\tau wr,$$

$$\sum_{w,r}c_{w,r}=\beta\sum_{w,r}wQ_{w,r}+\gamma\sum_{w,r}Q_{w,r}+\tau\sum_{w,r}wrQ_{w,r},$$

$$N=\beta+\gamma+\tau v.$$

Substituting, changing variables $\beta\leftarrow N\beta$ and $\tau\leftarrow N\tau$, and discarding constants yields

$$l_v(\beta,\tau)=\sum_n \log\left(1+\beta(w_n-1)+\tau(w_n r_n-v)\right).$$

## B  Derivation of Value Estimate

From equation (5),

$$\left\{v\middle| \sup_{(\beta,\tau)\in\mathcal{C}}l_v(\beta,\tau)=l^*_{\text{mle}}\right\},\tag{5}$$

we see any value estimate achieves the maximum dual likelihood value. Applying the duality established in Appendix A to $l_v(\beta,0)$ indicates all value estimates correspond to $v=\mathbb{E}_Q[wr]$ where $Q$ achieves the primal maximum

$$\sup_{Q\in\Delta}\left\{\sum_n \log\left(Q_{w_n,r_n}\right)\middle| \mathbb{E}_Q\left[w\right]=1\right\}.$$

Forming the Lagrangian dual

$$\sup_{\beta,\gamma} \inf_{Q \succeq 0} \left\{ \beta \left( -1 + \sum_{w,r} wQ_{w,r} \right) + \gamma \left( -1 + \sum_{w,r} Q_{w,r} \right) \right.$$
$$\left. - \sum_{w,r} c_{w,r} \log \left( Q_{w,r} \right) \right\},$$

where $c_{w,r} = \sum_n 1_{w=w_n, r=r_n}$. Collecting terms

$$\sup_{\beta,\gamma} \inf_{Q \succeq 0} \left\{ -\beta - \gamma \right.$$
$$\left. + \sum_{w,r} \left( \beta w + \gamma \right) Q_{w,r} - c_{w,r} \log \left( Q_{w,r} \right) \right\}.$$

Dual boundedness requires $\forall w : \beta w + \gamma \geq 0$. The infimum over $Q$ is separable yielding

$$Q^*_{w,r} = \frac{c_{w,r}}{\beta w + \gamma},$$

if $c_{w,r} > 0$ or $\beta w + \gamma > 0$, otherwise the contribution to the dual is zero. Substituting

$$\sup_{\beta,\gamma} \left\{ -\beta - \gamma + \sum_n \log \left( \beta w_n + \gamma \right) \middle| \forall w : \beta w + \gamma \geq 0 \right\}$$

discarding constants.

Summing the KKT stationarity conditions yields

$$\frac{c_{w,r}}{Q_{w,r}} = \beta w + \gamma,$$
$$\sum_{w,r} c_{w,r} = \beta \sum_{w,r} wQ_{w,r} + \gamma \sum_{w,r} Q_{w,r},$$
$$N = \beta + \gamma.$$

Substituting, changing variables $\beta \leftarrow N\beta$, and discarding constants yields

$$\sup_{\beta} \left\{ \sum_n \log \left( \beta(w_n - 1) + 1 \right) \middle| \forall w : \beta(w - 1) + 1 \geq 0 \right\}.$$

If $\beta^* = 0$ then $Q^*$ is supported only on the sample due to $1 = \mathbb{E}_Q[1]$. Otherwise, $Q^*$ is entirely supported on the sample except where $1 + \beta^*(w - 1) \geq 0$ is satisfied with equality. This can only be at the smallest or largest possible value of $w$ depending upon the sign of $\beta^*$; call this $w_{ex}$. Any $r$ is equally likely at this point; call it $\rho$.

Equation (6) follows via

$$\hat{V}(\pi) = \sum_{w,r} wQ_{w,r}r$$
$$= \sum_n w_n Q_{w_n,r_n} r_n + w_{ex} Q_{w_{ex},\rho} \rho$$
$$= \sum_n w_n Q_{w_n,r_n} r_n + \left( 1 - \sum_n w_n Q_{w_n,r_n} \right) \rho$$
$$= \rho + \sum_n w_n Q_{w_n,r_n} (r_n - \rho)$$
$$= \rho + \frac{1}{N} \sum_n \frac{w_n(r_n - \rho)}{1 + \beta^*(w_n - 1)}$$

where the first line is by definition, the third by $1 = \mathbb{E}_Q[w]$, and the fifth line by the primal-dual relationship.

# C Derivation of Lower Bound

The lower bound is the infimum of the value set defined by equation (8),

$$\left\{ v \;\middle|\; \sup_{(\beta,\tau)\in\mathcal{C}} l_v(\beta,\tau) - l^*_{\mathrm{mle}} \leq \frac{1}{2}\chi^{2,\alpha}_{(1)} \right\}. \tag{8}$$

Applying the duality established in Appendix A we get the equivalent primal formulation

$$\inf_{Q\in\Delta} \left\{ \mathbb{E}_Q[wr] \;\middle|\; \mathbb{E}_Q[w] = 1, \sum_n \log(Q_{w_n,r_n}) \geq \phi \right\}$$

where $\phi = \sum_n \log(Q^{\mathrm{mle}}_{w_n,r_n}) - \frac{1}{2}\chi^{2,\alpha}_{(1)}$. A Lagrangian dual is

$$\sup_{\substack{\kappa\geq 0 \\ \beta,\gamma}} \inf_{Q\succeq 0} \left\{ \beta\left(-1 + \sum_{w,r} wQ_{w,r}\right) + \gamma\left(-1 + \sum_{w,r} Q_{w,r}\right) \right.$$

$$\left. + \kappa\left(\phi - \sum_{w,r} c_{w,r}\log\left(Q_{w,r}\right)\right) + \sum_{w,r} wrQ_{w,r} \right\},$$

where $c_{w,r} = \sum_n 1_{w=w_n,r=r_n}$. Collecting terms

$$\sup_{\substack{\kappa\geq 0 \\ \beta,\gamma}} \inf_{Q\succeq 0} \left\{ -\beta - \gamma + \kappa\phi \right.$$

$$\left. + \sum_{w,r} \left(\beta w + \gamma + wr\right) Q_{w,r} - \kappa c_{w,r}\log\left(Q_{w,r}\right) \right\}.$$

Dual boundedness requires $\forall w,r : \beta w + \gamma + wr > 0 \vee (\beta w + \gamma + wr = 0 \wedge c_{w,r} = 0)$. The infimum over $Q$ is separable yielding

$$Q^*_{w,r} = \kappa\frac{c_{w,r}}{\beta w + \gamma + wr},$$

if $c_{w,r} > 0$ or $\beta w + \gamma + wr > 0$, otherwise the contribution to the dual is zero. Substituting and changing variables $\phi \leftarrow \frac{\phi-1}{N}$ yields

$$\sup_{\substack{\kappa\geq 0 \\ \beta,\gamma}} -\beta - \gamma + \sum_n \left( -\kappa\log\kappa + \kappa\big(\phi + 1 + \log\left(\gamma + \beta w_n + w_n r_n\right)\big) \right)$$

discarding constants.

$Q^*$ is supported on the sample except where $\beta w + \gamma + wr \geq 0$ is satisfied with equality. Because $wr \geq 0$, this implies equality can only happen at $wr = 0$ otherwise other violations occur. Thus all constraints are implied by $\forall w \in \{w_{\min}, w_{\max}\} : \beta w + \gamma \geq 0$. Denote $\Xi$ to be the set of $(w,r)$ pairs where equality occurs.

Equation (9) follows via

$$v_{\mathrm{lb}}(\pi) = \sum_{w,r} Q_{w,r} wr$$

$$= \frac{1}{N}\sum_n Q_{w_n,r_n} w_n r_n + \sum_{(w,r)\in\Xi} Q_{w,r} wr$$

$$= \frac{1}{N}\sum_n w_n Q_{w_n,r_n} r_n \qquad\qquad (\forall(w,r)\in\Xi : wr = 0)$$

$$= \kappa^* \frac{1}{N}\sum_n \frac{w_n r_n}{\gamma^* + \beta^* w_n + w_n r_n},$$

where the first line is by definition, and the fourth line by the primal-dual relationship.

## D  Proof of Theorem 1

**Lemma 1.** *Let $\beta^*$ solve*

$$\sup_{\beta}\left\{\sum_n \log\left(1+\beta(w_n-1)\right)\middle| \forall w : 1+\beta(w-1)\geq 0\right\}.$$

*Then*

$$|\beta^*|\sum_n \frac{(w_n-1)^2}{1+\beta^*(w_n-1)}\leq\left|\sum_n(w_n-1)\right|.$$

*Proof.* For the unconstrained maximizer,

$$0 = \sum_n \frac{w_n-1}{1+\beta^*(w_n-1)}$$

$$= \sum_n(w_n-1)\left(1-\frac{\beta^*(w_n-1)}{1+\beta^*(w_n-1)}\right),$$

$$\beta^*\sum_n \frac{(w_n-1)^2}{1+\beta^*(w_n-1)} = \sum_n(w_n-1),$$

$$|\beta^*|\sum_n \frac{(w_n-1)^2}{1+\beta^*(w_n-1)} = \left|\sum_n(w_n-1)\right|.$$

For the constrained maximizer, first note the sign of $\beta^*$ is the sign of $\sum_n(w_n-1)$ because $\beta=0$ is feasible and

$$\frac{\partial}{\partial\beta}\sum_n \log\left(1+\beta(w_n-1)\right)\bigg|_{\beta=0} = \sum_n(w_n-1).$$

If the constrained maximizer is positive than

$$0 < \frac{\partial}{\partial\beta}\sum_n \log\left(1+\beta(w_n-1)\right)\bigg|_{\beta=\beta^*}$$

$$= \sum_n \frac{w_n-1}{1+\beta^*(w_n-1)}$$

$$= \sum_n(w_n-1)\left(1-\frac{\beta^*(w_n-1)}{1+\beta^*(w_n-1)}\right),$$

$$\beta^*\sum_n \frac{(w_n-1)^2}{1+\beta^*(w_n-1)} < \sum_n(w_n-1),$$

$$|\beta^*|\sum_n \frac{(w_n-1)^2}{1+\beta^*(w_n-1)} < \left|\sum_n(w_n-1)\right|.$$

If the constrained maximizer is negative than

$$0 > \frac{\partial}{\partial\beta}\sum_n \log\left(1+\beta(w_n-1)\right)\bigg|_{\beta=\beta^*}$$

$$= \sum_n \frac{w_n-1}{1+\beta^*(w_n-1)}$$

$$= \sum_n(w_n-1)\left(1-\frac{\beta^*(w_n-1)}{1+\beta^*(w_n-1)}\right),$$

$$\beta^* \sum_n \frac{(w_n - 1)^2}{1 + \beta^*(w_n - 1)} > \sum_n (w_n - 1),$$

$$|\beta^*| \sum_n \frac{(w_n - 1)^2}{1 + \beta^*(w_n - 1)} < \left| \sum_n (w_n - 1) \right|.$$

. $\qquad\qquad\qquad\qquad\qquad\qquad\qquad\qquad\qquad\qquad\qquad\qquad\qquad\qquad$ $\square$

**Lemma 2.** *Let $\{\sum_{k \leq n}(w_k - 1)\}_{n \in N}$ be a martingale sequence adapted to the filtration $\{\mathcal{F}_n\}_{n \in N}$ where a.s. $\forall n : 0 \leq w_n \leq w_{\max}$ with $w_{\max} \geq 1$. Then*

$$\mathbb{E}\left[ \frac{1}{N} \left| \sum_{n \leq N}(w_n - 1) \right| \right] \leq 5\sqrt{\frac{y}{N}} + 8\frac{w_{\max}}{N},$$

*where a.s.*

$$y \geq \frac{1}{N} \sum_{n \leq N} \mathbb{E}\left[ (w_n - 1)^2 | \mathcal{F}_n \right].$$

*Proof.* Freedman's inequality indicates

$$\Pr\left( |M_N| \geq x, \langle M \rangle_N \leq y \right) \leq 2 \exp\left( -\frac{x^2}{2(y + w_{\max}x)} \right),$$

where $M_N \doteq \sum_n \Delta M_n$, $\Delta M_n \doteq w_n - 1$, $\langle M \rangle_N \doteq \sum_{n \leq N} \mathbb{E}\left[ \Delta M_n^2 | \mathcal{F}_{n-1} \right]$. Let $\mathcal{E}$ denote the event $\langle M \rangle_N \leq y$. Then

$$\mathbb{E}\left[ |M_N| 1_{\mathcal{E}} \right] = \int_0^\infty dx \, \Pr\left( |M_N| \geq x, \mathcal{E} \right).$$

We do the integration in pieces. For $x \geq \frac{y}{w_{\max}}$, we have

$$\int_{\frac{y}{w_{\max}}}^\infty dx \, \Pr\left( |M_N| \geq x, \mathcal{E} \right)$$

$$= \int_{\frac{y}{w_{\max}}}^\infty dx \, 2 \exp\left( -\frac{x^2}{2(y + w_{\max}x)} \right)$$

$$\leq \int_{\frac{y}{w_{\max}}}^\infty dx \, 2 \exp\left( -\frac{x}{4w_{\max}} \right)$$

$$= 8w_{\max} \exp\left( -\frac{y}{4w_{\max}^2} \right)$$

$$\leq 8w_{\max}.$$

Therefore

$$\mathbb{E}\left[ |M_N| 1_{\mathcal{E}} \right]$$

$$\leq 8w_{\max} + \int_0^{\frac{y}{w_{\max}}} 2 \exp\left( -\frac{x^2}{2(y + w_{\max}x)} \right)$$

$$\leq 8w_{\max} + a + \int_a^{\frac{y}{w_{\max}}} 2 \exp\left( -\frac{x^2}{2(y + w_{\max}x)} \right)$$

$$\leq 8w_{\max} + a + \int_a^{\frac{y}{w_{\max}}} 2 \exp\left( -\frac{x^2}{4y} \right)$$

$$\leq 8w_{\max} + a + 2\sqrt{\pi}\sqrt{y}\left( 1 - \text{erf}\left( \frac{a}{2\sqrt{y}} \right) \right)$$

$$\leq 8w_{\max} + 2\sqrt{y}\left( \sqrt{\pi}\left( 1 + \text{erf}\left( \sqrt{\log(2)} \right) \right) - \sqrt{\log(2)} \right)$$

$$\leq 5\sqrt{y} + 8w_{\max}.$$

Dividing by $N$ completes the proof. $\qquad\qquad\qquad\qquad\qquad\qquad\qquad\qquad\qquad$ $\square$

**Theorem 1.** *Let* $\hat{R}(\pi) \doteq \frac{1}{N} \sum_n \frac{(w_n - 1)(r_n - \rho)}{1 + \beta^*(w_n - 1)}$ *with* $\beta^*$ *as in eq. (7), and let a.s.* $\forall n : 0 \leq w_n \leq w_{\max}$ *with* $w_{\max} \geq 1$. *Then*

$$\left| \mathbb{E}\left[\hat{R}(\pi)\right] - R(\pi) \right| \leq 10\sqrt{\frac{w_{\max}}{N}} + 16\frac{w_{\max}}{N}$$

*where* $R(\pi) \doteq V(\pi) - V(h)$ *is the true policy value difference between* $\pi$ *and* $\{h_n\}_{n \in N}$.

*Proof.* Consider the random variable

$$\Delta\hat{R}(\pi) = \hat{R}(\pi) - \frac{1}{N}\sum_n (w_n - 1)(r_n - \rho)$$

$$= \frac{1}{N}\sum_n \frac{\beta^*(w_n - 1)^2}{1 + \beta^*(w_n - 1)}(r_n - \rho).$$

$\Delta\hat{R}(\pi)$ is the difference of $\hat{R}(\pi)$ and an unbiased estimator, therefore its expectation is the bias of $\hat{R}(\pi)$.

$$\left| \mathbb{E}\left[\Delta\hat{R}(\pi)\right] \right|$$

$$\leq \mathbb{E}\left[\left|\Delta\hat{R}(\pi)\right|\right]$$

$$\leq 2\mathbb{E}\left[\frac{1}{N}|\beta^*|\sum_n \frac{(w_n - 1)^2}{1 + \beta^*(w_n - 1)}\right]$$

$$\leq 2\mathbb{E}\left[\frac{1}{N}\left|\sum_n (w_n - 1)\right|\right]$$

$$\leq 10\sqrt{\frac{y}{N}} + 16\frac{w_{\max}}{N}.$$

Finally we can bound $y$ via $\mathbb{E}[(w_n - 1)^2 | \mathcal{F}_{n-1}] \leq \mathbb{E}[w_n^2 | \mathcal{F}_{n-1}] \leq w_{\max}\mathbb{E}[w_n | \mathcal{F}_{n-1}] \leq w_{\max}$. □

## E   Off-Policy Evaluation, Synthetic Data

First, an environment is sampled. For all environments, the historical logging policy is $\epsilon$-greedy with possible importance weights $(0, 2, 1000)$. We choose $\pi$ to induce the maximum entropy distribution over importance weights consistent with $\mathbb{E}[w^2] = 100$. Rewards are binary with the conditional distribution of reward varying per environment draw such that the value of $\pi$ is uniformly distributed on $[0, 1]$. Once an environment is drawn a set of examples is sampled from that environment, and the squared error of the value estimate is computed.

## F   Off-Policy Evaluation, Realistic Data

We use the following 40 datasets from OpenML [31] identified by their OpenML dataset id: 1216, 1217, 1218, 1233, 1235, 1236, 1237, 1238, 1241, 1242, 1412, 1413, 1441, 1442, 1443, 1444, 1449, 1451, 1453, 1454, 1455, 1457, 1459, 1460, 1464, 1467, 1470, 1471, 1472, 1473, 1475, 1481, 1482, 1483, 1486, 1487, 1488, 1489, 1496, 1498. For each dataset we convert to Vowpal Wabbit format, shuffle the dataset, and utilize up to the first 10,000 examples as data. We utilize a 20%/60%/20% Initialize/Learn/Evaluate split sequentially by line number. Note the shuffle and split is done only once per dataset. We create a historical policy $h$ using on-policy learning on the Initialize dataset, and then learn a new policy $\pi$ on the Learn dataset using off-policy learning with data drawn from $h$. These Initialize and Learn steps are done once per dataset. Only the off-policy evaluation step is done multiple times per dataset, and the random variations are due to the different actions selected by $h$ over the Evaluate set. For each evaluation, we compute the squared error of the different predictors, i.e., the squared difference between the off-policy value estimate and the true value of $\pi$. Note the true value of $\pi$ can be computed (and is independent of the choices of $h$ on the evaluation set) because the underlying datasets are fully observed. Using the squared error as the random variable, we apply a paired $t$-test between EL and the other predictors to determine win, loss, or tie for each dataset. We use default settings for Vowpal Wabbit except for the choice of exploration strategy.

# G   Learning from Logged Bandit Feedback

We first utilize the same 40 datasets as above, but with a 20%/20%/60% Initialize/Learn/Evaluate split. The Initialize step is done once per dataset, then the Learn and Evaluate steps are done multiple times per dataset. Note the Evaluate step here is using the true value of $\pi$, i.e., is deterministic and independent of $h$ given $\pi$. Using the evaluation score as the random variable, we apply a paired $t$-test between MLE and the other predictors to determine win, loss, or tie for each dataset. We use Vowpal Wabbit in IPS learning mode with default settings, and do 4 passes over the data. At the beginning of each pass, we optimize the dual variables holding the policy fixed, then use the resulting dual variables during the learning pass to compute importance weights.

# H   Cressie-Read Divergence Results

We describe variants of the estimator and confidence interval utilizing the Cressie-Read power divergence, which takes the form

$$\mathrm{CR}(\lambda) = \frac{2}{\lambda(\lambda + 1)} \sum_n \left( (NQ_{w_n, r_n})^{-\lambda} - 1 \right)$$

with parameter $\lambda$. The choice $\lambda = -2$ is of practical interest because it yields closed-form solutions driven by sufficient statistics that are easily maintained online.

## H.1   Estimator

The primal formulation for the estimator is

$$\sup_{Q \in \Delta} \left\{ \sum_n \left( (NQ_{w_n, r_n})^2 - 1 \right) \middle| \mathbb{E}_Q[w] = 1 \right\}.$$

When optimizing over all distributions this can result in all the mass placed outside the sample, so we constrain the distributions to be supported on the empirical support plus an additional importance weight $w_{\mathrm{undata}}$, with arbitrary associated reward $\rho$, corresponding to where the KL divergence places additional support:

$$w_{\mathrm{undata}} = \begin{cases} w_{\min} & \frac{1}{N} \sum_n w_n \geq 1 \\ w_{\max} & \text{otherwise} \end{cases}.$$

This results in closed form solution

$$Q_{w,r} = -\frac{\gamma^* + \beta^* w}{2(N + 1)},$$

where

$$\begin{pmatrix} \gamma^* \\ \beta^* \end{pmatrix} = \begin{pmatrix} \frac{b}{b - a^2} \\ -\frac{a}{b - a^2} \end{pmatrix},$$

$$a \doteq \frac{1}{N + 1} \sum_{n \cup \{\mathrm{undata}\}} (w_n - 1),$$

$$b \doteq \frac{1}{N + 1} \sum_{n \cup \{\mathrm{undata}\}} (w_n - 1)^2.$$

The resulting value estimate interval is

$$\hat{V}(\pi) = \rho + \frac{1}{N} \sum_n \left( \left( \frac{N}{1 + N} \right) \gamma^* w_n + \left( \frac{N}{1 + N} \right) \beta^* (w_n - 1)^2 \right) (r_n - \rho)$$

where $\rho \in [0, 1]$. Sufficient statistics for the estimator are $N$ and the (unaugmented support) empirical sums of $w$ and $w^2$.

## H.2 Confidence Interval

The primal formulation for the lower bound is

$$\inf_{Q \in \Delta} \left\{ \mathbb{E}_Q[wr] \Big| \mathbb{E}_Q[w] = 1, \sum_n \left( N Q_{w_n, r_n} \right)^2 - 1 \geq \phi \right\}$$

where $\phi = \frac{1}{2} \chi_{(1)}^{2,\alpha} - \left( \sum_n \left( N Q_{w_n, r_n}^{\text{mle}} \right)^2 - 1 \right)$.

When optimizing over all distributions this can result in all the mass placed outside the sample, so we constrain the distributions to be supported on the empirical support plus an additional importance weight and reward pair. We consider both extreme points $\{(w, 0) | w \in \{w_{\min}, w_{\max}\}\}$ corresponding to where the KL divergence might place additional support, and use the minimum value as the lower bound. This results in a closed-form solution

$$Q_{w,r} = -\frac{\gamma^* + \beta^* w + wr}{(N+1)\kappa^*},$$

where

$$\begin{pmatrix} \gamma^* \\ \beta^* \end{pmatrix} = \kappa^* \vec{a} + \vec{b},$$

$$\vec{a} \doteq \frac{1}{\overline{w^2} - \overline{w}^2} \begin{pmatrix} -\overline{w^2} & \overline{w} \\ \overline{w} & -1 \end{pmatrix} \vec{1},$$

$$\vec{b} \doteq \frac{1}{\overline{w^2} - \overline{w}^2} \begin{pmatrix} -\overline{w^2} & \overline{w} \\ \overline{w} & -1 \end{pmatrix} \begin{pmatrix} \overline{wr} \\ \overline{w^2 r} \end{pmatrix},$$

$$x \doteq \overline{wr} + \frac{(1 - \overline{w})\left( \overline{w^2 r} - \overline{w}\,\overline{wr} \right)}{\overline{w^2} - \overline{w}^2},$$

$$y \doteq \frac{\left( \overline{w^2 r} - \overline{w}\,\overline{wr} \right)^2}{\overline{w^2} - \overline{w}^2} - \left( \overline{w^2 r^2} - \overline{wr}^2 \right),$$

$$z \doteq \phi + \frac{(1 - \overline{w})^2}{2\left( \overline{w^2} - (\overline{w})^2 \right)},$$

$$\kappa^* = \sqrt{\frac{y}{2z}},$$

where $\overline{(\cdot)}$ denotes empirical mean including augmented support. The resulting lower bound is $v_{\text{lb}}(\pi) = x - \sqrt{2yz}$.

Sufficient statistics for the lower bound are $N$ and the (unaugmented support) empirical sums of $w$, $w^2$, $wr$, $w^2 r$, and $w^2 r^2$.