[Reviews · NeurIPS 2020]

Review 1

Summary and Contributions: This work tackles the problem of providing confidence intervals on the estimates of a contextual bandit policy learned offline. More specifically, this paper shows that one can use the empirical likelihood (EL) to efficiently compute confidence intervals that are tighter than the ones obtained by others methods while simultaneously working in regimes where others would fail. The authors also provide a bound on the bias on the resulting estimator and empirically highlight the performance of the approach.

Strengths: The investigated approach (EL) for computing the confidence intervals in the offline contextual bandit policy evaluation setting seems relevant for the task. The resulting estimator seems reasonable to use in practice (is tractable) and is proven to have low bias. Several empirical experiments are provided to illustrate properties of the approach. The papers also draws connections with other estimators (e.g. doubly-robust). The tools provided in this paper could be useful for analyzing techniques that would rely on such EL estimator.

Weaknesses: Due to a lack of contextual information (e.g. the problem is not even clearly stated), this work is hard to follow, which strongly undermines the potential impact of the paper. It is not clear what each experiment is investigating, what claims are supported by the presented results, how these results should be interpreted. It is also unclear how novel this work is: it is presented as a specialization of the estimator proposed in [Kallus and Uehara, 2019] and it is not obvious how much novelty this implies, i.e. how the use of EL differs in the current case and how impactful the analysis of EL under the current setting could be.

Correctness: Given what can be understood from the paper, the claims seem sound. The empirical methodology in the experiments appears to be correct.

Clarity: The writing is ok but very dry. The paper lacks the contextual information to situate the reader properly in order to understand the work.

Relation to Prior Work: It is not clear how EL applied to offline contextual bandits differs from EL applied to robust supervised learning. It is also unclear how regularizing counterfactual learning via lower-bound optimization in the current setting differs from previous applications.

Reproducibility: Yes

Additional Feedback: This seems like interesting work but it would need more context to be fully appreciated. More specifically, the introduction could be improved to better situate the reader regarding the problem and setting that are being considered in this paper; in the current state, it assumes too much prior knowledge from the reader. ============== Update ================ Reading other reviews and the rebuttal have clarified some points. Moreover, I have realized that the paper might be more impactful than what I initially thought. I have adjusted my score accordingly.


Review 2

Summary and Contributions: This paper proposes a new CI method for off-policy estimation based on dual likelihood. The obtained coverage is asymptotically nominal and empirically performs well on finite data. Computing the interval and the corresponding point estimate requires solving a convex optimization problem and is fairly efficient.

Strengths: This paper builds on the dual likelihood method (Mykland,‎1995), which as far as I am aware of, has not yet received attention in the setting of off-policy estimation. The dual likelihood statistic has been previously shown to achieve nominal coverage for non-iid data, which is essential for this application. The key contribution of this work is to introduce the dual likelihood technique to off-policy estimation (which to me is not an obvious step). Computational concerns have been addressed as well. Further, the authors show that a certain bias term is well-controlled, and a link to distributionally robust optimization is pointed out. While perhaps most theoretical results used in the submission have been known before, phrasing these in the off-policy estimation setting is a significant conceptual contribution in my opinion. I liked that pedagogical example. Empirical evidence is provided as well.

Weaknesses: As presented, the method is limited to bounded rewards. Further it is unclear to me if the support of the distribution over (w,r) needs to be finite. Please clarify. It would be great if the pedagogical example could be extended to the dual likelihood method as well.

Correctness: The theoretical claims and the methodology appears correct to me.

Clarity: Overall the paper is well organized; but there are a few points that need improvement. When I was reading the paper for the first time, many ideas appeared very confusing to me (perhaps the author's could also get someone else's feedback regarding this for an updated version). In particular, the concepts introduced in 3.1. and 3.2 (eq (3) and (5)) were not easy to grasp for me. Some more explanation and intuition would be very welcome. Also, the relation between MLE, EL and (5),(6) should be clarified. In particular, Figure 2 only shows MLE (and not EL) and it is unclear to me if these are equivalent to (5). - Some abbreviations are not introduced when used first, e.g. "IPS" (line 49) "EMP" (line 128) - line 33 "optimizes over distributions": It is unclear at that point what these distributions are. - I couldn't find the distribution "Q" introduced formally except for in the pedagogical example. It is used beyond that, in particular in section 3.3., and therefore needs a formal definition. - line 92 the claim "behaves as a likelihood" needs more explanation. - line 105: it is unclear how eq (7) defines the value l_{mle}^* - line 118: Consider adding \rho as an argument/subscript to \hat V(\pi) to make clear that \rho needs to be chosen.

Relation to Prior Work: In my view, related work is adequately discussed.

Reproducibility: Yes

Additional Feedback: line 10: what is meant by "lower (data) scale"? line 36: "intuitively centered" -> very informal line 39: "pleasing functional form" -> very informal UPDATE: I have read the author's feedback and will keep my score.


Review 3

Summary and Contributions: This paper considered the problem of estimating the average reward and its confidence interval of applying any policy $\pi$ in a contextual bandit problem, given a series of data generated from some other learning policies. They proposed an estimation method called empirical likelihood, which is efficient and performs well in experiments.

Strengths: The estimation of the average reward and its confidence interval of applying an arbitrary policy $\pi$ in a contextual bandit problem from a series of data generated from some other learning policies is an important problem in contextual bandit, especially when the support size of contexts are infinite. This paper provide a novel method to do so, which is both effective and efficient in experiments. This makes their empirical likelihood method very attractive.

Weaknesses: I do not find any theoretical guarantee for the estimation of confidence intervals. Do you have any theorems about the error probability of your estimated confidence intervals? When you choose to build an $\alpha$-confidence interval, is the real error probability smaller than $\alpha$ theoretically?

Correctness: I check most of the proofs in the supplementary file, and I think they are correct.

Clarity: The paper is well written.

Relation to Prior Work: It is clearly discussed.

Reproducibility: Yes

Additional Feedback: I am wondering that whether this method can be extended to a more general setting, i.e., to estimate the average reward of applying some policy in an online reinforcement learning problem. ==============After read other reviews and the rebuttal================ My final evaluation will not change.


Review 4

Summary and Contributions: The authors consider the contextual bandit problem and propose a novel, nonparametric approach based on empirical likelihoods. This new, flexible method results in narrow confidence sets with good coverage as demonstrated in Figure 1. The authors propose a convex optimization method for computing the confidence intervals improving upon standard bisection methods. They also derive finite sample bounds for the bias of the centering point of the confidence interval. Finally, after relating their work to the literature, they demonstrate good finite sample empirical properties of their approach on a simulationsstudy.

Strengths: The article considers an important and popular topic and actually I am a bit surprised that so far nonparametric approaches weren't proposed in the literature for constructing confidence intervals. The derived results are interesting and perhaps even more importantly seem to improve the state of the art results, especially for uncertainty quantification.

Weaknesses: I didn't find major limitations, beside that it could be made more accessible for non-specialised audience, see my detailed comments below. It would be nice to give guarantees for the size of the confidence interval and also the mse of the estimator not just the bias, but I understand that this might go beyond the scope fo the paper. I have a minor question though: I did not understand why in line 55 the authors said that w_n can have only 3 possible values? If I understood correctly the authors only assumed that \pi is discrete...

Correctness: The claims and the methodology seem to be correct.

Clarity: The level of the English is very good, however, I found the paper a bit hard to read. I understand that the paper is highly technical, and the authors deliver a lot, but it is clearly written for experts. In my opinion it would be helpful for an interested reader without deep knowledge in the literature to provide more explanation for the key results for instance by removing less important results to the supplement. This also holds for the simulation study. Furthermore I have found several typos and have a few minor comments: - l 54: r should be bold - l 62-62: dot is missing in display. - l 105: referring forward to eq (7) is not always the best idea - l 146: Alteratively -> Alternatively - l 272: rephrase the second part of the sentence (grammar) ================= Update =================== My opinion did not change about the paper, I would still advise the authors to improve the readability.

Relation to Prior Work: As far as I know the authors discuss the literature well.

Reproducibility: Yes

Additional Feedback:

[Author Response · NeurIPS 2020]

We thank the reviewers for constructive feedback.

Multiple reviewers indicated the set of primal distributions being optimized over is not clearly articulated. Given
the dataset, we optimize over the set of distributions which dominate the empirical distribution (i.e., place positive
probability on each realized datum). Despite this being a broad class, the support of the optimum matches the empirical
support plus one additional $(w, r)$ tuple (cf. Sections 4, B, and C). We've adjusted the exposition to indicate this.

Detailed responses to concrete questions or comments follow. (numbers in brackets refer to references in the paper)

**Reviewer 1**: *it is presented as a specialization of . . . [Kallus and Uehara, 2019]*: This is not correct—our approach is
an alternative. The estimators agree with an asymptotically large number of samples, but disagree with fewer samples
as shown in Figure 2. Furthermore this approach enables our primary focus on CIs for both off-policy estimation and
robust off-policy learning in contextual bandits. CIs are not addressed in [Kallus and Uehara, 2019].

*what each experiment is investigating*: We adjusted the beginning of section 6 to introduce the experiments. Experiments
investigate the quality of estimation, CIs, and learning. In estimation we compare MSE of different estimators. In CIs
we compare coverage and width of different CIs. In learning we compare accuracy of learned policies with training
objective our CI lower bound (or estimator) against the algorithm used in the VW system, a mature software for
contextual bandit learning.

*the relationship to robust supervised learning*: In Section 5 we refer the reader to [7] for how empirical likelihood
applies to supervised learning. For contextual bandits one needs to account for the nature of the partial feedback and
cannot simply use the formulation from [7]. The relationship then is that both [7] and our work propose learning a
model under the worst distribution that is still plausible given the data.

**Reviewer 2**: *Does the support of the distribution over $(w, r)$ need to be finite?* No. The asymptotic distribution of
the dual likelihood statistic is due to a martingale CLT [21]; bounded moments are sufficient and finite support is not
required.

*Figure 2 using MLE instead of EL*: fixed. Furthermore in the text we now consistently refer to equation (6) as "EL".

*Introduce abbreviations when first used*: fixed for IPS. EMP is the actual term used by [Kallus and Uehara, 2019].

*behaves as a likelihood*: Dual likelihood ratios are asymptotically distributed like parametric likelihood ratios in the
well-specified case [21], providing a nonparametric analogue to Wilks' theorem. We've clarified the exposition.

*unclear how eq(7) defines the value*: We now discuss the "maximum possible dual likelihood value given the data" and
refer the reader to section 3.2.

*what is meant by lower (data) scale*: adjusted to read "the amount of data required for success"

*intuitively centered*: dropped. *pleasing functional form*: "pleasing" dropped. all other comments: fixed.

**Reviewer 3**: *I do not find any theoretical guarantee for the estimation of confidence intervals*. The CIs have correct
asymptotic coverage and coverage errors decay as $O(1/n)$ (cf. [23] section 2.6). We now indicate this explicitly.

*When you . . . build an $\alpha$-confidence interval, is the real error probability smaller than $\alpha$ theoretically?*: This is an open
question. Empirical evidence (Fig. 1 right) suggests we do.

*whether this method can be extended to a more general setting*: we are currently researching this. Unfortunately, due to
space constraints, we had to drop discussion of follow-on research in section 7.

**Reviewer 4**: *it could be made more accessible for non-specialised audience*. We tried to include enough background,
pointers to relevant work, and an example in Section 2.

*give guarantees for the size of the confidence interval and also the mse of the estimator*. Our estimator asymptotically
coincides with [Kallus and Uehara 2019] and the asymptotic MSE was derived in that paper. For the CI size the results
of [14] (also [23] Section 13.5) show that EL enjoys a kind of optimality similar to that of the likelihood ratio test for
multinomial samples [10].

$w_n$ *can only have 3 possible values*: the logging policy $h$ is $\epsilon$-greedy and the evaluated policy $\pi$ is deterministic so the
3 possible values correspond to $h$ and $\pi$ disagree, or they agree and $h$ explores or exploits. We now state this explicitly.

*referring forward to equation (7)*. We now discuss the "maximum possible dual likelihood value given the data" and
refer the reader to section 3.2.

all other comments: fixed.

[Meta-Review · NeurIPS 2020]

This submission investigates the use of the empirical likehood approach for off-line policy evaluation in contextual models (which, by the way, is not made fully explicit from its title). The paper was considered as a novel and conceptually interesting contribution by all reviewers who recommend its acceptance. I agree with this general opinion, with the added warning that both the reviews and the post-rebuttal reviewers' discussion made it clear that the current writting needs improvement in several ways. Please note that this a unanimous request of the reviewers (even those that gave very good rating), as these quotes from the discussion will easily show: "After reading the other referee reports and the authors comments I still have the feeling that the authors could have done a better job in formulating the problems and giving enough context (providing a bunch of references is not equal to writing an easy to read, accessible and good paper). This paper was by far the hardest to read for me this year [...]", "I also found the paper really hard to read, up to the point where I found that this was severely undermining its potential impact.", "I agree with what has been said so far. I hope that the feedback will lead to some improvements to the presentation of the results." Going in the same direction I can also add that I share the same concerns, despite the fact that I knew what the empirical likelihood method was, before reading the paper. The authors in their answer committed to do the requested changes when preparing the final version of the paper, and in the present case it is very important that they take this occasion to improve the general presentation of the paper.